# Measurement report: In-situ vertical profiles of below-cloud aerosol over the central Greenland Ice Sheet.

Heather Guy[1,2], Andrew S. Martin[1,2], Erik Olson[3], Ian M. Brooks[2], and Ryan R. Neely III[1,2]

[1]National Centre for Atmospheric Science, Leeds, U.K.
[2]School of Earth and Environment, University of Leeds, Leeds, U.K.
[3]Space Science and Engineering Center, UW-Madison, Madison, WI, USA

**Correspondence:** Heather Guy (heather.guy@ncas.ac.uk)

**Abstract.**

Surface radiative cooling in polar regions can generate persistent stability in the atmospheric boundary layer. Stable layers below clouds can decouple the cloud layer from the near-surface environment. Under these conditions, surface aerosol measurements are not necessarily representative of the near-cloud or intra-cloud aerosol populations. To better understand the variability in the vertical structure of aerosol properties over the central Greenland Ice Sheet, in-situ measurements of aerosol particle size distributions up to cloud base were made at Summit Station in July and August 2023. These measurements identified distinct vertical aerosol layers between the surface and cloud base associated thermodynamic decoupling layers. Such decoupling layers occur 49% of the time during the summer in central Greenland, suggesting that surface aerosol measurements are insufficient to describe the cloud-relevant aerosol population half of the time. Experience during this first measurement season demonstrated the ability of a tethered balloon platform to operate effectively under icing conditions and at low surface pressure ($< 680$ hPa). The results presented here illustrate the value of vertically resolved in-situ measurements of aerosol properties to develop a nuanced understanding of the aerosol effects on cloud properties in polar regions.

## 1 Introduction

Clouds are an important control on the surface energy budget of the Greenland Ice Sheet (GrIS). Clouds increase downwelling longwave radiation relative to equivalent clear sky conditions and, in the summer, they shade the surface from incoming solar radiation. The net radiative effect of clouds at the surface depends on the incoming solar radiation and surface albedo as well as the temperature, optical depth, and microphysical properties of the cloud (Shupe and Intrieri, 2004). As a result, changes in cloud cover over the GrIS can have either a net warming or a net cooling effect at the surface, which is both regionally and seasonally dependent. Van Tricht et al. (2016) demonstrated that, on annual timescales, the longwave warming effect of clouds prevails, leading to an overall warming of the ice sheet. Conversely, Hofer et al. (2017) found that a reduction in summer cloud cover enhances net downwelling radiation over the lower albedo ablation zone, resulting in increased surface melt. Cloud radiative forcing is particularly sensitive to the integrated amount of cloud liquid water (liquid water path, LWP, e.g. Miller et al., 2015). Bennartz et al. (2013) showed that LWP was a critical control on surface melt in central Greenland during the

extreme melt event in July 2012. Understanding the processes that control LWP and cloud lifetime over the GrIS is essential
for understanding how the GrIS surface energy budget will respond to changes in cloud cover.

Aerosol particles are an important control on cloud lifetime and phase, and hence LWP. The abundance of ice nucleating
particles (INPs) in a supercooled cloud can determine how much ice forms. When the air is supersaturated with respect to ice
but subsaturated with respect to water, ice particles will grow at the expense of liquid droplets, reducing LWP and longwave
cloud radiative forcing (e.g.Korolev, 2007). Since ice particles tend to be larger than liquid droplets, ice formation controlled
by the abundance of INPs can impact cloud lifetime (e.g. Storelvmo et al., 2011). The abundance of particles that can nucleate
liquid cloud droplets (cloud condensation nuclei, CCN), controls the number of cloud droplets that form at a given supersatu-
ration. This means that a cloud with the same liquid water content, but fewer CCN, will consist of fewer, larger cloud droplets
(Twomey, 1977). Such a cloud would be less opaque than the equivalent cloud with more CCN, increasing shortwave trans-
mittance. In extreme cases, a lack of CCN can trigger a positive feedback where the few activated droplets grow large enough
to precipitate out, removing any remaining CCN and limiting cloud LWP, longwave emissivity, and cloud lifetime (Mauritsen
et al., 2011; Sterzinger et al., 2022).

Despite the potential sensitivity of cloud properties, and therefore cloud radiative forcing, to the number concentrations of
CCN and INP, measurements of aerosol properties over the GrIS are sparse (Schmale et al., 2021). Year-round surface-based
measurements of particle number concentrations and size distributions, which are important for determining the concentrations
of CCN and INP, are only available from Villum Research Station (north-east Greenland) since 2010 and from Summit Station
(central GrIS) since 2019 (fig. 1). Villum, like other coastal Arctic sites, is sensitive to Arctic haze and marine aerosol sources
(Nguyen et al., 2016; Freud et al., 2017; Lange et al., 2018), whereas Summit is sensitive to the descent of aerosol particles from
the free troposphere (Hirdman et al., 2010; Law et al., 2014; Guy et al., 2021). Since the primary source of aerosol particles at
Summit is descent from above, surface measurements are strongly impacted by fog scavenging and air mass isolation below
near-surface temperature inversions, implying that they might not be representative of the aerosol population higher in the
atmosphere (Dibb et al., 1992; Bergin et al., 1994, 1995; Guy et al., 2021, 2023).

Near-surface temperature inversions, formed by radiative cooling of high emissivity snow- and ice-covered surfaces in the
Arctic, can act to thermodynamically decouple the surface from the upper boundary layer (Shupe et al., 2013; Sotiropoulou
et al., 2014; Brooks et al., 2017). Where such decoupling occurs, distinct differences between the surface and cloud-relevant
aerosol populations have been directly observed (Igel et al., 2017; Creamean et al., 2021; Lonardi et al., 2022; Zhang et al.,
2022). Creamean et al. (2021) use equivalent potential temperature profiles (a measure of static stability), alongside vertically
resolved measurements of particle number concentrations from Oliktok Point in Alaska, to show that whether or not the surface
aerosol population is similar to that at cloud base is tightly coupled to the thermodynamic mixing state of the boundary layer.

Over central Greenland, near-surface temperature inversions occur over 70% of the time (Hoch et al., 2007; Miller et al.,
2013), and the high static stability within the inversion inhibits vertical mixing. Despite the importance of understanding the
cloud-relevant aerosol population over the GrIS, and the fact that this population is likely to differ from that at the surface
due to the high near-surface static stability, measurements of aerosol particles at cloud height over the GrIS are limited to
just a handful of aircraft campaigns (Flyger et al., 1973, 1976; Law et al., 2014). Although useful for understanding aerosol

properties at cloud height, the spatial and temporal resolution of aircraft measurements are severely limited, and aircraft are
unable to take repeated measurements close to the surface to understand the relationship between surface aerosol properties,
which are relatively straightforward to measure, and those at cloud height.

Here we demonstrate the use of a tethered balloon platform to measure vertically resolved aerosol particle size distributions
up to 830 m a.g.l over the central GrIS. Although limited to just six opportunistic sampling days in July and August of 2023,
the measurements demonstrate the utility of this measurement platform in the cold, remote, and high-altitude environment
of central Greenland, paving the way for future campaigns of longer duration that are necessary to understand the vertical
structure of near-surface aerosol particles over the GrIS, and their relevance for radiatively important cloud properties.

## 2 Measurements and methodology

### 2.1 Sampling location

Summit Station (72.58°N, -38.45°E) is in the accumulation zone of the GrIS, on the summit plateau 3,250 m above mean
sea level (fig. 1). There are no local sources of primary aerosol particles apart from station emissions. As an atmospheric
baseline sampling site, non-essential emissions are strictly controlled to protect the quality of long-term trace gas measurements
collected at the Atmospheric Watch Observatory (AWO) south of the main station. The base station for vertical aerosol profiling
was located ∼2,600 m west of the main station generator (fig. 1). The potential for contamination of the aerosol measurements
by pollution from the main station generator is discussed in section 4. As part of the station-wide emission control protocol, we
were not permitted to travel (via snowmobile) to our sampling location when the wind direction was between 285° and 39°.
Unfortunately, despite the statistical unlikelihood, 10 of 14 possible sampling days were impacted by this restriction which
significantly limited the number of profiles we were able to sample during this campaign.

### 2.2 Measurement platform

Vertical profiling was enabled by a 21 m$^3$ Helikite (fig. 2). The Helikite is a tethered helium filled balloon with a kite wing that
provides orientation, stabilisation, and additional dynamic lift (Allsopp Helikites Ltd., 2023). At sea level pressure, the balloon
envelope provides 12 kg of static lift. At Summit Station, where the mean surface air pressure in summer is 680 hPa, the static
lift is reduced to 5 kg. In winds of 4.2 m s$^{-1}$, the kite wing increases the lift by ∼50%, according to the manufacturer.

The Helikite was attached to a 1 km tether (2.5 mm diameter line weighing 4.9 g m$^{-1}$), selected to provide the maximum
sampling altitude whilst allowing sufficient lift for the instrument package at full extension. The tether was passed through an
anchored redirection point at the base station. The other end of the tether was attached to a snowmobile which was used to
raise and lower the measurement platform.

The main advantages of the Helikite platform over drone or aircraft platforms are that it can sample through clouds, in icing
conditions, and can profile in the same location for extended periods of time. Ice forming on the Helikite and tether may cause
the Helikite to descend gradually, but not result in a sudden uncontrolled loss of altitude. The Helikite can nominally operate in

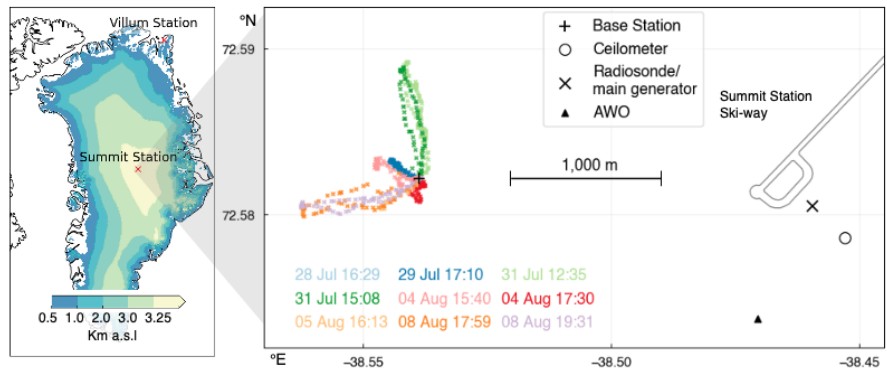

**Figure 1.** Location and map of Summit Station. Coloured makers show the location of the measurement package during sampling with relation to the tethered balloon base station on the ground, which is also indicative of the wind direction during each profile. The surface based POPS was located at the Atmospheric Watch Observatory (AWO) south of the main station. Ice elevation contours are from the Greenland Ice Mapping Project (Howat et al., 2017).

wind speeds of 0 to 18 m s$^{-1}$, although in practice, managing the Helikite on the surface was challenging when winds exceeded 7 m s$^{-1}$.

## 2.3 Instrumentation

The Helikite instrument package consisted of a Handix Portable Optical Particle Spectrometer (POPS Gao et al., 2016; Mei et al., 2020), and a reusable S1H3 Windsond radiosonde. The combined instrument payload including battery pack weighed
3 kg. The POPS nominally measures size-resolved aerosol particle number concentrations in 16 bins from 115 to 3370 nm diameter, however several studies have found that the POPS tends to over-count at small particles sizes due to stray light in the optical chamber (Gao et al., 2016; Pilz et al., 2022; Pohorsky et al., 2024). For this reason, and because of high percentage uncertainties in particle counts in the smallest two size bins (>50 %, Handix Scientific, 2022b), we discard the smallest two size bins and only consider particles between 136 and 3370 nm diameter.

The POPS deployed on the Helikite (henceforth the sky-POPS) was placed in a lightweight insulating foam box, and a coarse mesh filter was placed over the inlet to prevent the growth of rime ice. The inlet was otherwise as provided by the manufacturer and the air was not dried prior to sampling, hence all size-resolved POPS measurements in this study are referring to the wet particle diameter. The sky-POPS was secured to the kite wing such that the inlet was always oriented into the wind (fig. 2). The sky-POPS was factory calibrated prior to the deployment using PSL spheres (Handix Scientific, 2022b). The PSL spheres
had a refractive index of 1.61. Since information about aerosol particle composition at Summit Station is extremely limited, no correction for refractive index was applied to the POPS data.

A second POPS instrument has been operating at the AWO since August 2022, and is henceforth referred to as the AWO-POPS. This instrument measures ambient air through an omnidirectional inlet and was also factory calibrated with PSL spheres

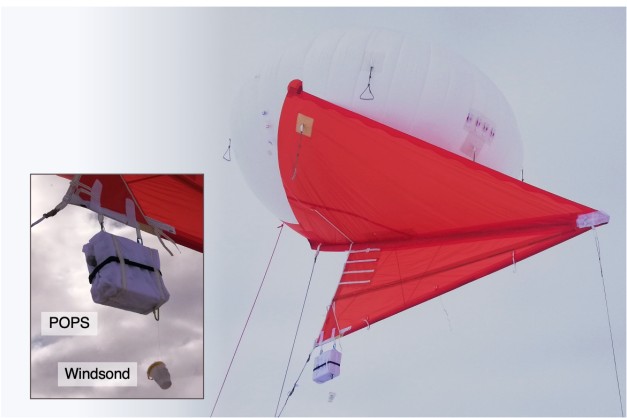

**Figure 2.** Inflated Helikite and instrument payload.

prior to deployment (Handix Scientific, 2022a). A bench top intercomparison and zero-check of the two instruments was carried
out at Summit on 22 July 2023. In the absence of an independent reference instrument, this intercomparison was intended to
allow for a relative calibration between the two instruments. Since the AWO-POPS has been operating at Summit for a full
year, it may have experienced some calibration drift that we have not quantified here. For the zero check, a 0.01 micron
borosilicate glass microfiber filter was attached to the inlet of both instruments. The standard deviation of the noise in the total
particle number concentration during the zero check was 0.23 cm$^{-3}$ for the sky-POPS and 0.02 cm$^{-3}$ for the AWO-POPS.
The AWO-POPS consistently undercounted the total particle number concentration (136 to 3370 nm wet diameter) relative to
the sky-POPS by a factor of 0.66 $\pm$0.17 (1 s.d.). For the purpose of this study we are only interested in the relative changes
in the aerosol size distribution as a function of height, so we apply a bias correction to the AWO-POPS total particle number
concentration by dividing the measured concentration by 0.66. The addition of the mesh filter to the sky-POPS inlet did not
significantly impact the sky-POPS sampling efficiency relative to the AWO-POPS. Size dependent uncertainties in the aerosol
particle size distribution and total aerosol number concentrations measured by each POPS instrument were calculated from
the combined uncertainty from the manufacturers sampling efficiency estimation and flow-rate calibration, and the noise from
the onsite zero check. Refer to Gao et al. (2016); Mei et al. (2020); Pilz et al. (2022) and Pohorsky et al. (2024) for further
descriptions of the characteristic uncertainties associated with POPS measurements of particle size distributions.

The Windsond radiosonde was attached to the Helikite just beneath the POPS (fig. 2). The Windsond included an integrated
GPS unit and measured air pressure, temperature, and relative humidity at 1 Hz with accuracies of $\pm$1 hPa, 0.3 °C, and 2%
respectively. Windsond data were transmitted in real-time to the base station.

The Windsond has an automated algorithm that corrects the temperature and humidity measurements for the impact of
solar heating. This algorithm assumes a vertical ascent rate of > 2 m s$^{-1}$ and is not appropriate for our tethered balloon
measurements. We therefore applied the following quality control procedure to the raw data to remove data points that may
have been impacted by solar heating:

1. Removal of measurements collected when the measurement platform was moving slower than 0.5 m s$^{-1}$.

2. Application of a despiking algorithm to remove data points when the sensor was re-equilibrating. This algorithm identifies and removes points that lie outside of one standard deviation from the 50 m rolling mean of the temperature profile and is applied three times.

3. Manual removal of any remaining suspect data points. This included the lowest 30 m for all profiles where it was unclear if the sensor was equilibrated with ambient conditions after being stationary at the surface.

The threshold values used in this algorithm (0.5 m s$^{-1}$ and three repeats for the despiking algorithm) were determined by visually inspecting the raw data from the temperature profile measured on the day with the slowest wind speeds (case (e), see description in section 2.4). This case featured distinct increases in the raw temperature data when the Helikite was held 140 stationary at 50 m vertical intervals. Our quality control algorithm is therefore conservative for the rest of the case studies that took place under increased horizontal wind speeds (and therefore had greater sensor ventilation). The same quality control is also applied to the relative humidity measurements (the relative humidity sensor has the same thermal response time as the temperature sensor). After the quality control algorithm a 20 m rolling mean was applied to the good data points (consistent with the Windsond manufacturer's algorithm). Equivalent potential temperature ($\theta_e$) profiles were calculated from the temperature, 145 humidity, and pressure profiles using the MetPy python package (May et al., 2024).

### 2.4 Case studies

Conditions were suitable for operating the tethered balloon on 13 out of 14 potential sampling days from 25 July to 09 August 2023 (on 09 August the wind speed was too high, >10 m s$^{-1}$). Unfortunately, due to restrictions on snowmobile usage under certain wind directions (described in 2.1), we were only able to collect nine vertical profiles over six days (Table 1). We focused 150 on sampling the vertical aerosol profile below cloud base, although sampling through cloud would be possible during future campaigns with appropriate flying permissions. Horizontally extensive low-level stratocumulus or broken altocumulus, with cloud base ranging from 300 to 1,700 m a.g.l, was present on each sampling day. The sampled air pressure ranged from 617 to 684 hPa, and air temperature between -13.5 and -3.7 °C. The maximum 10 m wind speed during tethered balloon operations was 6.8 m s$^{-1}$. The maximum sampling altitude of 830 m a.g.l was achieved at full tether extension on a low wind day (04 155 August) when there was insufficient lift to fly higher. When wind speeds were faster, the maximum flying altitude was limited by the smaller angle of inclination between the tether and the ground. Repeat profiles were collected on the three days we were able to sample above 500 m a.g.l (31 July, 04, and 08 August).

### 2.5 Cloud and boundary layer structure

Additional information on the temporal evolution of cloud properties and boundary layer structure during the campaign are 160 available from a Vaisala CT25K ceilometer and twice daily launches of Vaisala RS41 radiosondes (at 00 and 12 UTC) as part of the 'Integrated Characterisation of Energy, Clouds, Atmospheric properties, and Precipitation at Summit' project (ICECAPS,

**Table 1.** Sampling times and associated atmospheric conditions. Altitude, air pressure, temperature, and relative humidity are from the Windsond radiosonde. Wind speed and direction (10 m) during each sampling period are from the NOAA GML meteorological station (NOAA-GML, 2023), and mean cloud base height is from the ICECAPS ceilometer (Shupe, 2010). Sample profiles are organised so that the lower altitude profiles are labelled (a)-(c) and the higher profiles (d)-(f).

| ID | Date DD/MM 2023 | Sampling time start/end (UTC) | Max profile altitude (m a.g.l) | Air pressure Max/min (hPa) | Air temperature Max/min °C | Relative humidity w.r.t. water Max/min | 10 m Wind speed Max/min (m s$^{-1}$) | 10 m Wind direction Mean (°) | Cloud base height median (m a.g.l) |
|---|---|---|---|---|---|---|---|---|---|
| (a) | 28/07 | 16:09/18:13 | 158 | 682/673 | -6.3/-7.4 | 79/73 | 6.7/4.3 | 108 | 315 |
| (b) | 29/07 | 17:00/19:00 | 262 | 678/659 | -7.5/-9.9 | 79/72 | 5.1/3.4 | 114 | 390 |
| (c) | 05/08 | 16:05/17:26 | 310 | 684/659 | -10.3/-12.4 | 78/69 | 3.3/1.8 | 19 | 1545 |
| (d) | 31/07 | 12:33/14:39 | 636 | 676/622 | -7.4/-13.5 | 73/60 | 3.1/1.2 | 198 | 1080 |
| (d$_1$) | 31/07 | 15:00/16:30 | 680 | 676/622 | -7.4/-13.5 | 77/64 | 2.4/0.4 | 205 | 1050 |
| (e) | 04/08 | 14:35/17:09 | 824 | 679/617 | -3.7/-11.3 | 69/35 | 2.0/0.1 | 13 | 1650 |
| (e$_1$) | 04/08 | 17:25/18:46 | 830 | 679/617 | -5.0/-11.3 | 69/35 | 2.6/0.9 | 1 | 1680 |
| (f) | 08/08 | 17:56/19:15 | 636 | 684/634 | -9.4/-13.4 | 80/75 | 6.8/5.2 | 56 | 1035 |
| (f$_1$) | 08/08 | 19:27/20:47 | 618 | 684/636 | -9.8/-13.2 | 85/74 | 6.6/3.8 | 67 | 975 |

Shupe et al., 2013). The ceilometer measured range and sensitivity normalised backscatter (at 905 nm, see Münkel et al., 2007), and cloud base height. Although there were no direct precipitation measurements during the campaign, precipitation occurrence and timing on sampling days was logged by on-site observers.

To explore how the vertical structure of the aerosol particle size distribution is related to the thermodynamic structure of the boundary layer, averaged particle size distributions as a function of height for each profile (ascending and descending combined) were calculated by gridding the data onto regular 5 m vertical intervals and calculating a 20 m rolling mean (to match the resolution of the Windsond data). Thermodynamically stable 'decoupling layers', across which turbulent mixing is inhibited, were identified where $\theta_e$ increased monotonically with height for at least 20 m, following the methodology of Vüllers et al. (2021). Neutral / weakly stable layers were removed by applying the criterion that each decoupling layer must have a minimum 0.5 K increase in $\theta_e$. Individual layers were merged into a single layer if they were separated by less than 100 m vertically.

To contextualise the results of this study with respect to the longer term dataset of cloud and boundary layer structure at Summit, we also apply the same methodology to detect decoupling layers from 932 Vaisala radiosonde profiles launched at Summit during cloudy conditions in June, July, and August between 2010 and 2022 (Shupe and Walden, 2010). Where 'cloudy' profiles are identified from the positive detection of a cloud base height by the ceilometer (Shupe, 2010).

## 3 Results

### 3.1 Vertical aerosol profiles and sky condition

Figure 3 shows the measurement altitude and total particle number concentration (136 to 3370 nm) in relation to the ceilometer backscatter profile for each sampling day. In case (a), the Windsond failed at the highest point in the profile and there were no measurements during the descent. Note that the ceilometer clearly identifies cloud base height, but the signal may be attenuated within the cloud and therefore does not reliably show the cloud vertical extent. The overall range of total aerosol particle number concentrations was 14 to 55 cm$^{-3}$.

On the first two sampling days (fig. 3a and b, 28 and 29 July), the cloud base was less than 400 m, and there was little variation in the total particle number concentration with altitude below the cloud. However, on the latter four days, when the cloud base was higher (950 to 1,700 m), altitude-dependent variations in the particle number concentration below cloud base were apparent (fig. 3).

During case (c), we were unable to sample up to cloud base due to the implementation of a station wide restriction on snowmobile usage. Nevertheless, there was a clear difference between the particle number concentration near the surface ($<$ 40 cm$^{-3}$) and that above 250 m ($>$ 50 cm$^{-3}$). One hour prior to sampling, a precipitating stratus cloud was present at the approximate height of the increase in particle number concentration (fig. 3c). There was also fog early in the morning that day, which is known to contribute to the wet scavenging of aerosol particles in the surface mixed layer (Guy et al., 2021).

During case (d), it was snowing prior to sampling (between 04 and 10 UTC), from a cloud with a base height of ~300 m. During sampling, a cloud with base height of 1080 m thinned and began to break up, and observers noted intermittent light snow at the surface. Both profiles on this day showed an increase in particle number concentrations with height from ~30 cm$^{-3}$ near the surface to $>$ 50 cm$^{-3}$ above 300 m (fig. 3d).

Low wind speeds (0.1 to 2.6 m s$^{-1}$) during case (e) allowed us to manually raise and lower the tethered balloon without the use of a snowmobile. On this occasion, particle number concentrations were higher close to the surface than aloft. In both repeat profiles, there was a distinct, relatively low particle number concentration layer between 350 and 500 m (fig. 3e). Earlier in the day (between 04 and 12 UTC), it had been snowing heavily from a cloud layer at ~400 m.

The strong signal in the ceilometer backscatter during case (f) was caused by intermittent snow squalls beneath broken altocumulus cloud. Particle concentrations were relatively low compared to previous days (median value 22 cm$^{-3}$), but increased with altitude in both profiles (fig. 3f). On each of the three days when a repeat profile was possible, the vertical distribution of aerosol particles was similar between the two repeats (fig. 3d-f). For the rest of the figures in this paper, only the first of the two repeats are shown.

During all six case studies, the total particle number concentrations measured by the sky-POPS near the surface were in close agreement with those measured by the AWO-POPS (fig. 4). For cases (a) and (b), changes in the sky-POPS total particle number concentration during the profile closely align with changes measured at the surface by the AWO-POPS, suggesting that the particle number concentration was consistent throughout the sampled vertical profile (fig. 4a, b). In contrast, the changes in

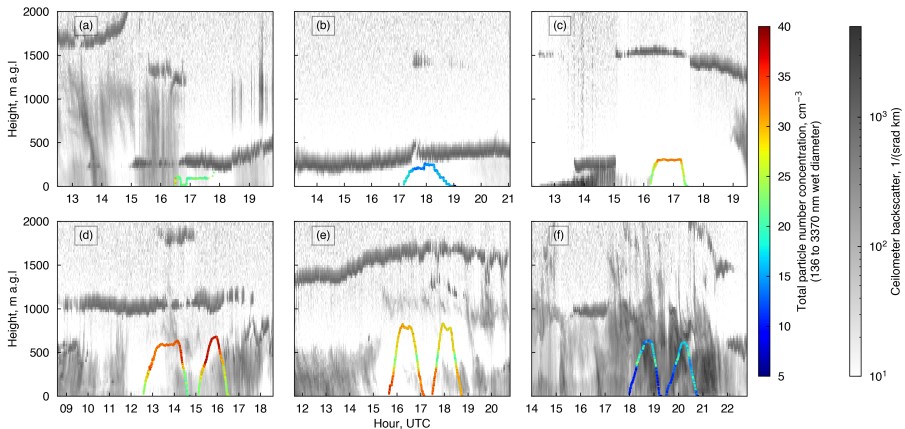

**Figure 3.** Vertical profiles overlain on ceilometer backscatter and coloured by total particle number concentration.

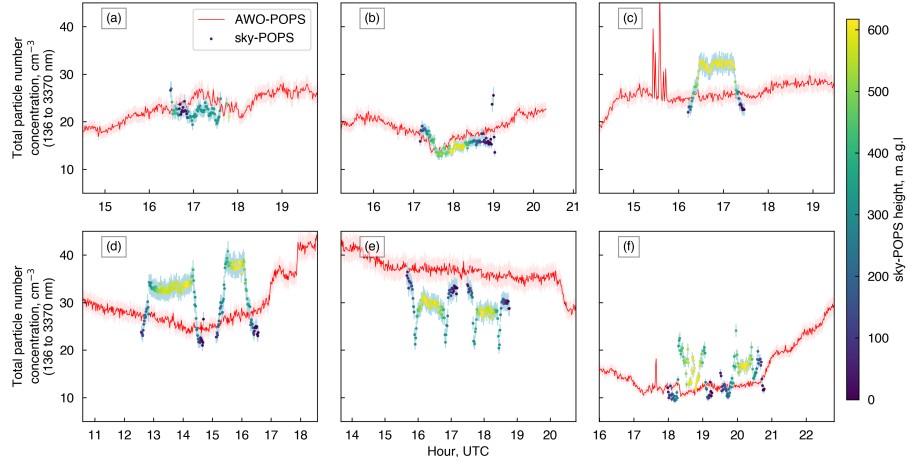

**Figure 4.** Total particle number concentration (136 to 3370 nm wet diameter) measured at the surface (AWO-POPS, red line) compared to the concentration measured during vertical profiling (sky-POPS, scatter points). Scatter points are colored to illustrate the sampling height of the sky-POPS). Note that a relative bias correction has been applied to the AWO-POPS measurements as described in the text. Absolute measurement uncertainties are shaded in red for the AWO-POPS and light blue for the sky-POPS.

the sky-POPS particle number concentrations during the profiles in cases (c)-(f) are not reflected in the surface observations, suggesting that these changes reflect variations in the vertical aerosol profile.

## 3.2 Relationship to boundary layer structure

On five of the six sampling days, the Windsond $\theta_e$ profile indicated that the cloud layer was thermodynamically decoupled from the surface. The mean height of the lower boundary of the decoupling layer was 241 m a.g.l (fig. 5). No $\theta_e$ profile is

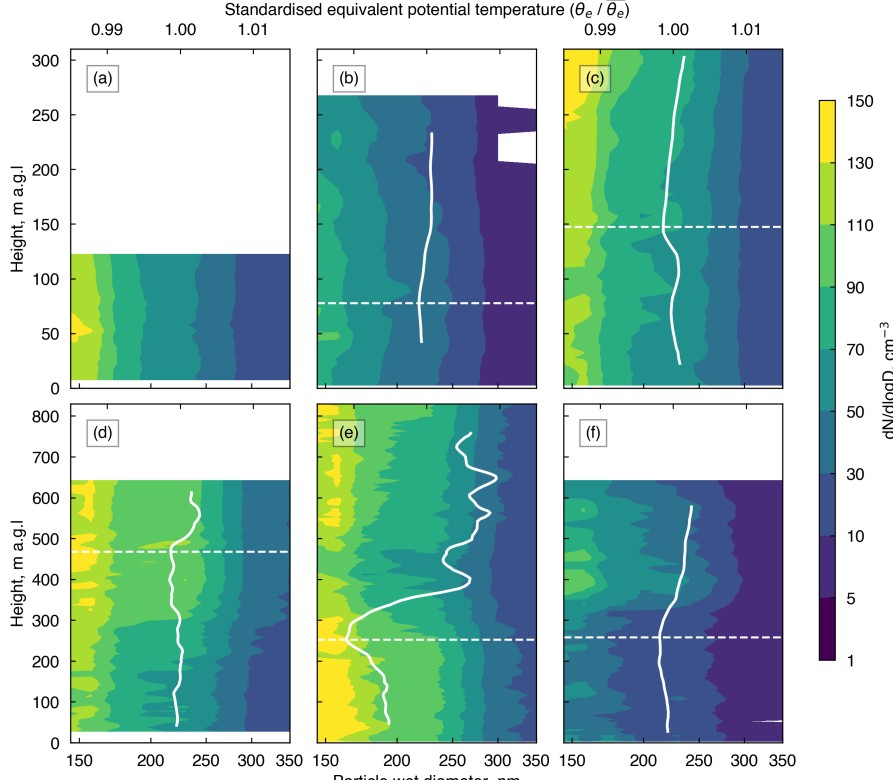

**Figure 5.** Particle size distributions (20 m rolling mean, contoured) and standardised equivalent potential temperature profiles ($\theta_e / \overline{\theta_e}$ , thick white line) for each profile. Where $\overline{\theta_e}$ is the mean value of the $\theta_e$ profile on that day. The three lower altitude flights are on the top row (note the difference in y-axis scales). Horizontal dashed lines indicate the lower boundaries of thermodynamically stable 'decoupling' layers.

shown for case (a) due to a Windsonde failure resulting in a lack of quality temperature and humidity measurements, however, the particle size distribution was well mixed throughout the profile (up to 130 m, fig. 5a). During case (b), the boundary layer was neutral up to a weakly stable (0.5 K) decoupling layer at 78-150 m. Above the decoupling layer, there were slightly fewer smaller particles (< 250 nm diameter) compared to within the surface mixed layer (fig. 5b, fig. 6b).

     During case (c), the near surface layer was well-mixed up to the start of a stable layer at 148 m (fig. 5c). On this occasion
there was a notable increase in smaller particles (< 300 nm diameter) above the surface mixed layer, with an 22% increase in particles smaller than 200 nm (fig. 6c). Fig. 5c shows that most of the increase in smaller particles occurred above ∼200 m (i.e. 50 m higher than the lower boundary of the stable decoupling layer).

     The decoupling layer identified for case (d) was located between 468 and 553 m a.g.l above a well-mixed surface layer (fig. 5d). On this day particle number concentrations increased with height above ∼250 m. The increase in particle number
concentrations between the surface mixed layer and above the decoupling layer is most pronounced for 160 to 250 nm diameter particles (fig. 5d, 6d).

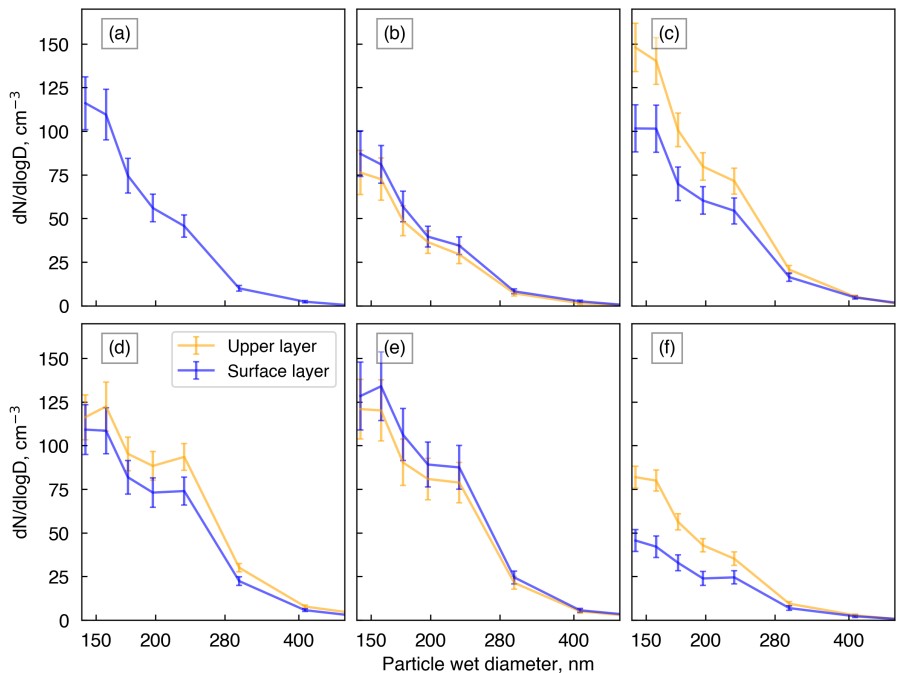

**Figure 6.** Mean particle size distribution in the surface mixed layer (blue) compared to the highest layer sampled (orange).

The surface layer was mostly well-mixed during case (e) up to a strong (6.3 K) decoupling layer with a base 253 m. Most of the inversion occurs in the first 150 m above that decoupling level, generating the steepest gradient in $\theta_e$ of the six cases (fig. 5e). The thermodynamically isolated surface mixed layer had the highest concentrations of 115 to 400 nm particles in the vertical profile of all the six cases (fig. 6e). Above the base of the decoupling layer there was a sharp decrease in particle number concentrations, particularly in the 140 to 280 nm diameter range. In all size bins, the lowest particle number concentrations occurred in a distinct layer between 350 and 500 m. At 500m, just above the strongest part of the decoupling layer, particle number concentrations increased again in all size bins and were then more or less constant with height for the rest of the profile (fig. 5e).

Finally, during case (f), the particle size distribution was constant with height throughout the well-mixed surface layer. At ~300 m, there was a sharp increase in the number concentration of 115 to 300 nm diameter particles, just above the lower boundary of a stable decoupling layer at 258 m (fig. 5f). Above this level the size distribution was approximately constant with height again, resulting in a clear distinction between the particle size distribution in the surface mixed layer and that in the below-cloud environment (fig. 6f).

Considering all the June, July, and August radiosonde profiles during cloudy times at Summit from 2010 to 2022 (932 radiosonde profiles in total), below-cloud thermodynamic decoupling layers occur 49% of the time. The average lowest decoupling height in this dataset was 120 m and 90% of decoupling heights were below 250 m.

## 4  Discussion

In summary, on five out of six sampling days, at least one thermodynamically stable layer was identified which would have inhibited the turbulent mixing of aerosol particles between the surface mixed layer and the below-cloud environment (fig 5). On the four of those days with the strongest decoupling layers, the total particle number concentration and size distribution varied as a function of height (figs. 4-6). On the day when there were insufficient measurements to calculate the $\theta_e$ profile (case a), the particle number concentration and size distribution was consistent with height up to cloud base, suggesting that this profile was well-mixed.

Of the cases where the surface mixed layer was thermodynamically decoupled from the cloud base, cases (b) and (e) had larger particle number concentrations within the surface mixed layer, whereas cases (c), (d), and (f), had larger particle number concentrations above the lower boundary of the decoupling layer. The larger particle number concentrations near the surface during case (e) may have resulted from a build-up of local station pollution during an extended period of low wind speeds; the average 10 m windspeed for the 24 hours prior the sampling was 1.9 m s$^{-1}$ ±0.6 (1 s.d.) Case (b) was the only day that was not impacted by precipitation either before or during sampling. The wet deposition of aerosol particles during precipitation events, either through in-cloud or below cloud scavenging, may have contributed to the depletion of aerosol particles in the isolated surface mixed layers during cases (c), (d), and (f), and to the depleted layer just above the decoupling height during case (e). During case (f) there was intermittent light snow throughout the day, and on the other days, there were heavier snow events preceding the measurement period. Fog in the early morning of case (c), which is known to deplete surface aerosol particles at Summit (Bergin et al., 1994, 1995; Guy et al., 2021), could also have contributed to the reduced particle concentrations in the surface mixed layer on this day.

Case (e) was unique in that a shallow layer that was depleted in aerosol particles occurred just above the lower boundary of the stable layer. Above the depleted layer particle number concentrations increased again (fig 5). This depleted layer was at the same altitude as a precipitating cloud that was present less than one hour before the the start of the measurement profile. One possible explanation for the depleted layer is that aerosol particles were scavenged within this cloud layer and subsequently removed by wet deposition.

The longer term dataset (2010-2022) of radiosonde launches at Summit demonstrate that the surface mixed layer is decoupled from the sub-cloud layer 49% of the time (Shupe and Walden, 2010), which is similar to observations from the central Arctic Ocean (Sotiropoulou et al., 2014; Brooks et al., 2017; Vüllers et al., 2021). This implies that during half of the summer period over the central Greenland Ice Sheet, when surface melt is significantly influenced by cloud properties (e.g. Bennartz et al., 2013), surface aerosol measurements do not accurately represent the aerosol population relevant to cloud interactions. Consequently, relying solely on surface aerosol measurements is inadequate for studying cloud-aerosol interactions in this context. In winter months, surface aerosol measurements are even less likely to be representative of the cloud relevant population, since persistent high static stability at the surface occurs over 80% of the time (Miller et al., 2013), and clouds with base heights < 2,000 m are less common (Shupe et al., 2013). This could explain the particularly low surface aerosol particle number concentrations at Summit Station during the winter (Guy et al., 2021).

During this measurement campaign, we only sampled below-cloud vertical aerosol profiles. The above-cloud aerosol population can also be an important source of cloud-relevant aerosol particles in the Arctic (Igel et al., 2017) and can vary significantly from the below-cloud environment (Igel et al., 2017; Creamean et al., 2021; Lonardi et al., 2022). Future measurement campaigns should aim to characterise the vertical aerosol distribution below, within, and above the cloud environment over Greenland.

Lightweight aerosol particle sensors and robust and versatile measurement platforms are improving our ability to collect longer-term measurements of vertical aerosol profiles in remote places. However, the expensive resource cost of these in-situ measurements (most notably in terms of person-power) means that they will always be of limited duration. To fully characterise the cloud-relevant aerosol population over longer periods and in all seasons, focus should be placed on using in-situ measurements to develop, calibrate, and evaluate ground-based remote sensing instrumentation that can detect the vertical structure of aerosol particles near to the surface and that can operate unattended year-round. For example, high spectral resolution lidar can separate molecular scattering and aerosol particle scattering signals to retrieve vertical profiles of aerosol scattering properties (e.g. Thorsen and Fu, 2015; Zhang et al., 2022). However, the relationship between aerosol scattering properties and cloud-relevant aerosol properties (i.e. particle size distribution or CCN concentration) varies depending on aerosol composition and shape (e.g. Ghan and Collins, 2004; Lv et al., 2018; Tan et al., 2019), and retrieving aerosol vertical profiles in the vicinity of clouds is complicated by precipitation and hygroscopic growth (e.g. Schmidt et al., 2014). Since lidar cannot detect aerosol particles through optically thick cloud, in-situ measurements remain the only way to sample the above cloud vertical aerosol profile.

## 5   Summary and conclusions

This report presents the first in-situ measurements of below-cloud vertically resolved aerosol particle size distributions over the central Greenland Ice Sheet. Although this campaign was limited to just nine vertical profiles on six opportunistic sampling days, the measurements demonstrate that surface-based aerosol measurements are not always representative of the cloud-relevant aerosol population over central Greenland. Thermodynamic decoupling between the surface and the cloud layer occurred on five of the six sampling days below horizontally extensive low-level cloud (cloud base 315 to 1,680 m), and distinct variations in the aerosol particle size distribution with height were associated with the decoupling layers in all but the weakest case. The fact that thermodynamic decoupling of the surface from cloud layers with bases < 2,000 m occurs 49% of the time during the summer in central Greenland suggests that surface aerosol measurements are insufficient to describe the cloud-relevant aerosol population half of the time. Given that the presence and LWP of low-level clouds are an important control on the ice sheet surface energy budget, and that the aerosol population can potentially modulate cloud LWP and lifetime, a concerted effort to understand aerosol vertical profiles and their importance for cloud-aerosol interactions over the Greenland Ice Sheet is warranted.

This measurement campaign demonstrates that a tethered balloon system carrying a simple optical particle counter can collect novel data about the vertical structure of the aerosol population in the lowest 800 m above the surface in extreme

conditions (freezing temperatures and low pressures) over the central Greenland Ice Sheet. Future campaigns should aim to sample both below and above cloud layers and should focus on characterising intra- and inter-seasonal variability and combining in-situ measurements with automated ground-based remote sensing to work towards the possibility of long-term measurements.

*Data availability.* All data collected during this measurement campaign are available at the CEDA data archive (Guy et al., 2024b, a, c). Complementary data from the ICECAPS project are available at the Arctic Data Center, at doi:10.18739/A20C4SM02 (ceilometer) and doi:10.18739/A2445HD3Q (radiosondes).

*Author contributions.* RN and HG conceptualised and planned the measurement campaign. AM, EO, and HG carried out the measurements in Greenland with offsite supervision from RN. HG led the data curation, analysis and visualisation with contributions from all co-authors. IB contributed to the analysis of the Windsonde data and boundary layer structure. HG wrote the manuscript with feedback from all co-authors.

*Competing interests.* The authors declare that they have no conflict of interest.

*Acknowledgements.* Thank you to staff and personnel at Summit Station during summer 2023 for exceptional support during the field campaign, and to Polar Field services for organisation and coordination of fieldwork activities. We are grateful to the Danish Civil Aviation Authority and Nuuk Flight Information Services for permission to fly the tethered balloon in the Northeast Greenland National Park. We acknowledge financial support from NSFGEO-NERC grants NE/X002403/1 and NE/S00906X/1. AM was also supported by the NERC SENSE-CDT NE/T00939X/1.

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
