# Peer review of "Measurement report: In-situ vertical profiles of below-cloud aerosol over the central Greenland Ice Sheet."

_EGUsphere, 2024_

## Author Comment (AC1)

**EGUSPHERE-2024-733: Author response to reviewer comments**

09 August 2024

**The co-authors would like to thank both reviewers for their feedback and thoughtful suggestions. We have responded to each comment below.**

Original reviewer comments are included in *italics.* Co-author responses are in simple text.

In addition to the changes specified in response to the reviewer comments, we have also made the following changes:

- Corrected dN /dlogD calculations, which previously had been calculated with respect to raw counts rather than concentrations.
- Added the position of the AWO onto figure 1.

**Response to RC1:**

*General comment*

> *Guy et al. present balloon-borne measurements of aerosol particle number concentration and size distribution in their manuscript at the Summit station in central Greenland. These observations are difficult to obtain and are a novelty in this region. On 6 days of operation, a helikite was used to carry an optical particle counter and a radiosonde to a maximum height of 830 m above ground level. The manuscript describes 6 aerosol profiles and puts them in context with atmospheric thermodynamic stability and de-coupling layers below the cloud base. Changing aerosol properties across the de-coupling layers indicate limited representativeness of surface-based aerosol measurements for cloud layers.*

> *However, the number of cases discussed in the manuscript is rather small, and the observations were limited to certain weather conditions that allow for balloon operations. Therefore, the presented manuscript seems insufficient to evaluate the broader relevance of the measurements. The study could benefit from setting the observations in context with other surface-based aerosol observations (in situ or remote sensing).*

We understand and agree with these significant limitations to this study. Our decision to publish these results as a 'measurement report' was largely due to the limited number of case studies and lack of supplementary ground-based observations that limit the scope of the conclusions we can draw. However, we believe that these results are useful to the scientific community despite their limitations due to the extreme challenges associated with collecting in-situ vertical measurements in remote polar regions. This is the first time that such measurements have been attempted in central Greenland, and we believe that this measurement report contains substantial useful information in both the methodology and results that will inform and contribute to the success of future more comprehensive campaigns.

> *The aerosol measurements with the POPS were neither validated, calibrated, or compared with other aerosol instruments. This is a significant flaw concerning reported measurement uncertainties of the POPS (see references below). The extensive post-processing of the radiosonde data from the helikite, including a subjective selection of valid or non-valid data, indicates unusual measurement uncertainties that require further validation by a reference.*

See specific comments on these points below.

*The aerosol measurements require more important detail: Which type of inlet was used? Please provide a particle size-dependent sampling efficiency estimation. Was the POPS calibrated or compared with a reference instrument?*

This was the first time this POPS had been used since it was provided with a calibration certificate from the manufacturer. No additional calibration or correction was applied. The POPS inlet is the same as the one provided by the manufacturer, and therefore the uncertainty analysis they provided is applicable (https://doi.org/10.5281/zenodo.11242687). The calibration was based on PSL spheres, and no correction has been made for refractive index since the refractive index of aerosol particles at Summit is unknown. The results of this calibration show that the POPS measurements were comparable to a reference CPC to within ~10% for particles greater than 136 nm diameter. For smaller particles, the manufacturers calibration suggests that the POPS undercounts relative to the CPC, this contrasts with studies that have found the POPS to overcount at small particles sizes due to stray light in the optical chamber (Gao et al., 2016; Pilz et al., 2022; Pohorsky et al., 2024). Due to this discrepancy and the higher uncertainties in the smaller size bins, we have chosen to discount particles smaller than 136 nm diameter from our analysis (the first two size bins), and have updated the figures accordingly.

On arrival at Summit Station, we ran a bench intercomparison between the POPS that was flown on the helikite (SN0307 henceforth sky-pops) and the POPS that has been operating at the Atmospheric Watch Observatory at Summit Station for the last two years (SN0288, henceforth awo-pops), which included a zero check for both instruments. In the absence of an independent reference instrument, this intercomparison was intended to allow for a relative calibration between the two instruments. Since the AWO-POPS has been operating at Summit for a full year, it may have experienced some calibration drift that we have not quantified here. For the zero check, a 0.01 micro borosilicate glass microfiber filter was attached to the inlet of both instruments. The standard deviation of the noise in the total particle number concentration was 0.23 $cm^{-3}$ for the sky-pops and 0.02 $cm^{-3}$ for the awo-pops. For the rest of the intercomparison the instruments sampled ambient air within the building. The coarse mesh filter added to prevent rime ice forming on the inlet did not impact the relative particle collection efficiency. The mean relative difference (awo-pops / sky-pops) in total number concentration was 0.66 +/- 0.17 (1 s.d) without the mesh filter, and 0.62 +/- 0.17 with the mesh filter installed. See the plots below (figures R1 and R2) for the size dependent differences between the two instruments. Overall, the sky-pops counted more particles than the awo-pops, particularly at particles sizes > 230 nm (bin 6). We have added error bars to figure 5 (now figure 6 in the revised paper) that illustrate the combined uncertainty from the manufacturers sampling efficiency estimation and flow-rate calibration, and the noise from the onsite zero check. We have also added these uncertainty estimations to the published data files.

[Figure]

**Figure R1:** Results of the benchtop intercomparison between the sky-POPS and AWO-POPS on 22 July 2023. Both instruments were sampling ambient air. Spikes in aerosol particle number concentrations were generated using velcro.

[Figure]

**Figure R2:** Relative difference in particle number concentration (AWO-POPS / sky-POPS) during the benchtop intercomparison. Solid line is the mean ratio and shaded regions are +/- 1 standard deviation.

To clarify these points and to provide the reader with a better understanding of the uncertainties associated with the POPS data, we have added more detail about the POPS calibration and benchtop intercomparisons in section 2.3, including the following paragraphs:

Lines 95 to 99:

"The POPS nominally measures size-resolved aerosol particle number concentrations in 16 bins from 115 to 3370 nm diameter, however several studies have found that the POPS tends to over-count at small particles sizes due to stray light in the optical chamber (Gao et al., 2016; Pilz et al., 2022; Pohorsky et al., 2024).
For this reason, and because of high percentage uncertainties in particle counts in the smallest two size bins (>50 %, Handix Scientific, 2022b), we discard the smallest two size bins and only consider particles between 136 and 3370 nm diameter."

Lines 101 to 102:

"The inlet was otherwise as provided by the manufacturer and the air was not dried prior to sampling, hence all size-resolved POPS measurements in this study are referring to the wet particle diameter."

Lines 104 to 106:

"The sky-POPS was factory calibrated prior to the deployment using PSL spheres (Handix Scientific, 2022b). Note that the PSL spheres had a refractive index of 1.61. Since information about aerosol particle composition at Summit Station is extremely limited, no correction for refractive index was applied to the POPS data."

Lines 107 to 123:

"A second POPS instrument has been operating at the AWO since August 2022, and is henceforth referred to as the AWO-POPS. This instrument measures ambient air through an omnidirectional inlet

and was also factory calibrated with PSL spheres prior to deployment (Handix Scientific, 2022a). A bench top intercomparison and zero-check of the two instruments was carried out at Summit on 22 July 2023. This intercomparison was intended to allow for a relative calibration between the two instruments rather than an absolute calibration. For the zero check, a 0.01 micron borosilicate glass microfiber filter was attached to the inlet of both instruments. The standard deviation of the noise in the total particle number concentration during the zero check was 0.23 $cm^{-3}$ for the sky-POPS and 0.02 $cm^{-3}$ for the AWO-POPS. The AWO-POPS consistently undercounted the total particle number concentration (136 to 3370 nm wet diameter) relative to the sky-POPS by a factor of 0.66 +/-0.17 (1 s.d.). For the purpose of this study we are only interested in the relative changes in the aerosol size distribution as a function of height, hence we apply a bias correction to the AWO-POPS total particle number concentration by dividing the measured concentration by 0.66. The addition of the mesh filter to the sky-POPS inlet did not significantly impact the sky-POPS sampling efficiency relative to the AWO-POPS. Size dependent uncertainties in the aerosol particle size distribution and total aerosol number concentrations measured by each POPS instrument were calculated from the combined uncertainty from the manufacturers sampling efficiency estimation and flow-rate calibration, and the noise from the onsite zero check. Refer to Gao et al. (2016); Mei et al. (2020); Pilz et al. (2022) and Pohorsky et al. (2024) for further descriptions of the characteristic uncertainties associated with POPS measurements of particle size distributions."

*Was the aerosol actively dried or did you measure wet particle diameter?*

The sample stream was not actively dried. Since the relative humidity during sampling did not exceed 75%, hygroscopic particle growth is likely to be small. We have clarified this in the text and specified 'wet' diameter in plot labels and in the published dataset.

Lines 101 to 102:

"The inlet was otherwise as provided by the manufacturer and the air was not dried prior to sampling, hence all size-resolved POPS measurements in this study are referring to the wet particle diameter."

*The POPS internal flow control system is well-known for increased uncertainties. Did you test the sample flow rate at the low ambient pressures at Summit?*

The POPS flow rate is monitored with a laminar flow element and differential pressure sensor that has been calibrated by the manufacturer (https://doi.org/10.5281/zenodo.11242687). According to Gao et al., (2016) there is no pressure dependency in this calibration down to 50 hPa. We did not measure the flow rate independently at Summit, but the flow rate measured by the LFE and differential pressure sensor during the measurement campaign ranged from 2.86 to 3.11 $cm^3$ $s^{-1}$ with a median flow rate of 2.98 +/- 0.01 (1 s.d.) $cm^3$ $s^{-1}$, which is within in the uncertainty range provided by the manufacturer. We have included the uncertainty in the flowrate (+/- 0.1 $cm^3$ $s^{-1}$ per the manufactures certificate) into the particle number concentration uncertainties that are now included in the published dataset.

*The POPS is known for increased measurement uncertainty towards smaller diameters, particularly below 150 nm. This can significantly reduce the useful size range of the POPS, e.g.*

*https://doi.org/10.5194/amt-17-601-2024*
*https://doi.org/10.5194/amt-17-731-2024*
*https://doi.org/10.5194/amt-15-6889-2022*

*Please provide a detailed evaluation of the POPS measurement performance including measurement uncertainties.*

As described above we have removed the first two size bins from our analysis and added error bars to figure 5 (now figure 6 in the revised paper) illustrating the combined uncertainty of the sampling efficiency estimation, flow-rate, and the noise level from the onsite zero check. These uncertainties are now also included in the published data file. We have added more information in section 2.3 about the uncertainty characterisation and limitations of the POPS measurements. See lines 95 to 123 in the revised text (also copied above).

*There seem to be too many post-processing steps for the Windsond data. How many raw data points per profile were left over before the final step 4? How did you validate the final data? How did it compare to the daily Vaisala radiosondes? Please provide a comparison of the raw data with the final post-processed data for each profile for the review process to allow for an evaluation of your post-processing routine.*

Figure R3 below illustrates the Windsond post-processing procedure for each of the (ascending) profiles. All the raw data points are shown by the grey (semi-transparent) points. Except for the first profile, we stepped the Helikite up in ~ 50 m increments, pausing for a couple of minutes to assess changing conditions between each increment. We dismissed all the Windsond data from the first profile due to inconsistencies in the GPS fix that resulted in intermittent data. The impact of the lack of ventilation on the sensor when the Helikite was stationary can be most clearly seen in panel (e), corresponding to the profile on 04th August, which had the lowest windspeeds of all the profiles (0.1 to 2 m s$^{-1}$ at 10 m). Similar features are also apparent in panels (c), (d) and (f), but less obvious in (b) when the cloud cover was thicker.

The Windsond manufacturer (Sparv) normally implement a correction algorithm to account for the impact of solar radiation on the sensor. Their proprietary algorithm takes into account the angle of the sun based on GPS coordinates, the air pressure reading, a light intensity measurement from the sensor, and implements a data smoothing algorithm over 20 m height intervals. This correction is not appropriate for our measurements as it assumes a vertical movement of 2 m s$^{-1}$ (manufacturer communication).

To generate a rigorous correction for solar heating during our measurements we would need to consider relative horizontal and vertical motion and the solar radiation intensity impacting the sensor. Unfortunately, the light intensity measurement from the sensor was not logged and we had no way to measure vertical profiles of horizontal wind speed. In the absence of these measurements or any independent method of validation, we implemented a quality control procedure based on visual examination of the data, starting by with the profile in panel (e), which had the most obvious artifacts. From this profile, we could see that the coolest points in the temperature profile (in-between the spikes in temperature when the Helikite was stationary) occurred when the Helikite was ascending at a speed of > 0.5 m s$^{-1}$. Since the horizontal wind speed was greater on all the other sampling days, the minimum ascent speed of 0.5 m s$^{-1}$ should be a conservative estimate. After filtering out all times when the ascent speed was less than this threshold, we were left with the red and cyan points in fig. R3. From these points there is clearly an equilibration time as the sensor returns to ambient conditions when the Helikite starts ascending. This equilibration time is dependent on the speed of the airflow past the sensor. In the absence of windspeed measurements, we choose to use a despiking algorithm to filter out the points where the sensor was still equilibrating. The parameters for the despiking algorithm (removal of points outside of one standard deviation from the 50 m rolling mean, and repeated three times), were selected based on visual inspection of the data in panel (e). Since we expect the equilibration time to be the slowest on profile (e) due to the lower surface wind speed, we expect this correction to be conservative for the other profiles. Some good data points are likely removed by the despiking algorithm, but not so many that the shape of the profile is meaningfully changed (fig. R3).

After the application of the despiking algorithm, each profile still had at least one good data point per 30 m (and usually more than that, see cyan points in fig. R3). The profile was then smoothed using a 20 m rolling mean in the vertical which is consistent with Sparv's usual smoothing algorithm.

The RH sensor is also affected by solar radiation, but the impacts are more difficult to see visually. Therefore, we only used the temperature profile to determine the 'good' data points, we then assume that when the temperature data are acceptable the RH data are also acceptable. This assumption is based on the fact that both sensors have the same thermal response time (manufacturer communication).

[Figure]

**Figure R3:** Vertical temperature profile measured by the Windsond during the first ascending profile for each case study. Grey semi-transparent markers are the raw Windsond measurement points. Cyan markers are the data points that pass our quality control procedure and are used to generate the final smoothed profile (black line). Red points are those that were removed by the despiking algorithm intended to remove points when the sensor was moving with a vertical velocity > 0.5 ms$^{-1}$ but was still equilibrating to the surroundings.

Unfortunately, none of the Vaisala radiosonde launches overlapped with the Windsond sampling times, and therefore we have no way to do an independent validation. Figure R4 illustrates how the quality controlled Windsond equivalent potential temperature profiles compare with the

Vaisala radiosonde profile that was launched closest in time to each Windsond profile (the launch times of each profile are included in the plot legends). For profiles (b), (c), (d), and (e), the equivalent potential temperature is consistently warmer near the surface in the Windsond profile compared to the Vaisala profile, but since the Vaisala launch was at ~ 10:00 local time and the Windsond profiles were typically just after local noon, this degree of warming is not unlikely. The difference in the shape of the Vaisala profile in profile (c) is related to fog that was present during the Vaisala launch which had dissipated before the Windsond launch. Figure 3 in the main text illustrates how much the cloud structure was changing with time for each profile. Qualitatively, the shape and magnitude of the Windsond equivalent potential temperature profile look reasonable compared to the Vaisala profiles given anticipated changes in the profile due to the diurnal solar heating and changes in cloud base and boundary layer structure.

[Figure]

**Figure R4:** Windsond equivalent potential temperature profiles for each case study (black lines) compared to the equivalent potential temperature profile from the Vaisala radiosonde that was launched closest in time to each Windsonde profile (red dashed line). The start times of each Windsond profile and the Vaisala radiosonde launch time are included in the legend inset (time are in UTC, local time is UTC-1).

We do not attempt to quantify any additional uncertainty in the temperature and humidity profiles related to the quality control algorithm, since in the absence of a method for independent validation, any attempt to do so would be subject to further assumptions. We do include quality control flags in the published dataset indicating suspect data points and points removed by the despiking algorithm. We believe that the shape of the equivalent potential temperature profiles that we use in our study to discuss boundary layer mixing are reliable for three reasons: (1) the similarity in the profile shape between the Windsond profiles and the Vaisala radiosonde profiles when temporal changes in cloud conditions and diurnal warming are considered, (2) the relationship between the height of features the thermodynamic profile and the height of features in the ceilometer data, and (3) the consistency in the shape of the profile between the 'good' data points identified with our quality control algorithm.

We have added more information describing the quality control procedure into section 2.3 to clarify the key points above:

Lines 127 to 145:

"The Windsond has an automated algorithm that corrects the temperature and humidity measurements for the impact of
solar heating. This algorithm assumes a vertical ascent rate of $> 2$ m s$^{-1}$ and is not appropriate for our tethered balloon
measurements. We therefore applied the following quality control procedure to the raw data to remove data points that may
have been impacted by solar heating:

1. Removal of measurements collected when the measurement platform was moving slower than 0.5 m s$^{-1}$.
2. Application of a despiking algorithm to remove data points when the sensor was re-equilibrating. This algorithm identifies and removes points that lie outside of one standard deviation from the 50 m rolling mean of the temperature profile and is applied three times.
3. Manual removal of any remaining suspect data points. This included the lowest 30 m for all profiles where it was unclear if the sensor was equilibrated with ambient conditions after being stationary at the surface.

The threshold values used in this algorithm (0.5 m s$^{-1}$ and three repeats for the despiking algorithm) were determined by
visually inspecting the raw data from the temperature profile measured on the day with the slowest wind speeds (case (e),
see description in section 2.4). This case featured distinct increases in the raw temperature data when the Helikite was held
stationary at 50 m vertical intervals. Our quality control algorithm is therefore conservative for the rest of the case studies that took place under increased horizontal wind speeds (and therefore had greater sensor ventilation). The same quality control is
also applied to the relative humidity measurements (the relative humidity sensor has the same thermal response time as the
temperature sensor). After the quality control algorithm, a 20 m rolling mean was applied to the good data points (consistent
with the usual Windsond algorithm). Equivalent potential temperature profiles were calculated from the temperature, humidity,
and pressure profiles using the Metpy python package (May et al., 2024)."

*Specific comments:*

*Table1: Which radiosonde, the windsond on the helikite or the Vaisala?*

The Windsond on the helikite. This is now explicit in the caption.

*L 97: The presented uncertainties seem unrealistic compared to the extensive post-processing. Please derive a realistic measurement uncertainty for RH and T from the raw data set.*

The quoted uncertainties are those of the raw sensor measurements and do not account for uncertainties related to solar heating/ insufficient sensor ventilation. As described above, we have not attempted to quantify any additional uncertainty that may have been introduced by solar heating due to the absence of a method for independent validation. Since we did not apply any additional correction to the raw data points that were flagged as 'good', the sensor uncertainties are still appropriate for these measurements, the only additional source of uncertainty would be if some of the data points flagged as 'good' were still impacted by the solar heating. As described above, we believe that our quality control algorithm is conservative and therefore maintain that the quoted uncertainties are our best estimates of the uncertainties in the 'good' data from the Windsond profile.

*L 137:A minimum theta E increase of 0.1 K seems lower than the measurement uncertainty of the windsond for RH and T. Please provide an error propagation to justify this criterion*

The threshold of 0.1 K was selected to be consistent with the methodology in previous studies (Vüllers et al., 2021). Propagating the uncertainties in the Windsond measurements (0.3 K for temperature, 2% relative humidity and 1 hPa for pressure) through the calculation of $\theta_e$, the absolute uncertainty in $\theta_e$ for the range of temperatures, pressures and RH values in this study ranges from 0.15 to 0.49 K. The uncertainty in the point-to-point differences in $\theta_e$ in the vertical is less than this since the profiles have been smoothed and measurements are correlated in the vertical domain. However, to be conservative, we have increased the threshold for the required increase in $\theta_e$ indicative of a decoupling layer to be at least 0.5 K. The primary impacts this has on our results is removing the lower weak decoupling layer that had previously been identified during case (d) and increasing the altitude of the decoupling layer in case (c). We have propagated this change throughout the text and updated Figures 4 and 5 (now figures 5 and 6 in the updated manuscript) accordingly. Our conclusions are not affected. Note that we used the MetPy library to do the $\theta_e$ calculations and have referenced this in the data files and in the text.

*L 145 to 147: The total particle numbers need to be validated. Pilz et al. 2022 (https://doi.org/10.5194/amt-15-6889-2022) found that the POPS can show high noise in terms of false particle concentrations up to 10 cm$^{-3}$ during measurements of particle-free air. Your measurements could be highly biased by the POPS uncertainties.*

As described above we have added information about the expected uncertainties based on the manufacturer calibration sampling efficiency estimation and the noise level derived from the on-site zero check.

*Figure 4: Please provide further info how the standardized equivalent pot. Temperature profiles were calculated*

The equivalent potential temperature profiles were calculated using the python MetPy library from the Windsond temperature, pressure, and relative humidity measurements. We have added a reference to this library in line 145 and in the data files. The standardised equivalent potential temperature profiles plotted in fig. 4 (now figure 5) are simply the $\theta_e$ profile divided by the mean value of the $\theta_e$ profile for each case. Standardising the profile in this way allows for an easier visual comparison between the strength of the stable layers in each case. We have updated the figure caption to clarify that the mean $\theta_e$ is the mean for the individual profile.

*The base of the de-coupling layer in panel (c) appears to be rather at 150 m and the upper one in panel (d) at 500 m. Please provide the raw temperature profiles from the windsond for evaluation (only for review)*

The change that we made to the threshold for identifying the decoupling layers per your earlier query about the measurement uncertainty in $\theta_e$ has impacted the level of the decoupling layers slightly. Both of the decoupling layers identified in (c) and (d) are now in agreement with your qualitative assessment. The raw temperature profiles from the Windsond are included above for your reference.

*Figure 5: All size distributions appear to be biased by increased noise in the bins below 150 nm, please check this with a zero filter. Also, please provide error bars or similar to indicate measurement uncertainties within the plot. Update the x-axis caption to wet diameter if the wet diameter was measured.*

We have added error bars to this figure (now figure 6) as described above and added 'wet diameter' to the caption. Based on the zero check that was performed prior to sampling, we did not observe increased noise in the smaller size bins. Nonetheless, due to larger uncertainties associated with the first two size bins and the results of previous studies as mentioned above, that attribute higher noise in the smaller size bins to stray light in the optical chamber, we have now removed the first two size bins (particles < 136 nm diameter) from our analysis.

*L213: Please provide reference measurements from the nearby station to validate this assumption*

We have added the following clarification: Line 254 : " the average 10 m windspeed for the 24 hours prior the sampling was 1.9 ms$^{-1}$ +/- 0.6 (1 s.d.)"

*L214: Please specify what you mean by nucleation and how it contributes to a depletion in aerosol particles.*

We were referring to the wet deposition of aerosol particles due to the nucleation of particles into cloud droplets which then freeze and precipitate, removing the aerosol particles from the cloud layer. We have modified this sentence to specifically refer to wet deposition:

Line 255-257: "The wet deposition of aerosol particles during precipitation events, either through in-cloud or below cloud scavenging, may have contributed to the depletion of aerosol particles in the isolated surface layers during cases (c), (d), and (f), and to the depleted layer just above the decoupling height during case (e)."

*L225: Please explain in more detail how the depleted layer was potentially affected by a cloud. No cloud is visible on the ceilometer during the balloon flight. Horizontal inhomogeneity of clouds/atmospheric layers and temporal evolution is hard to disentangle with the provided information*

We have extended the time axis in figure 3 by one hour so that the period of strong reflectively in the ceilometer data that occurred just prior to the Helikite profile and at the same altitude as the depleted aerosol layer is clearer. However, we appreciate that we do not have sufficient information to disentangle horizontal inhomogeneity in the boundary layer structure from temporal evolution. The fact that the depleted aerosol layer is at the same altitude as the precipitating cloud layer that was present less than one hour prior suggests that the two features may be related, and one hypothesis to explain this feature is that in-cloud scavenging of aerosol particles occurred in this cloud layer followed by wet deposition. To make it clearer that this is just one plausible hypothesis, we have changed this text to the following:

Line 263-266: "This depleted layer was at the same altitude as a precipitating cloud that was present less than one hour before the start of the measurement profile. One possible explanation for the depleted layer is that aerosol particles were scavenged within this cloud layer and subsequently removed by wet deposition."

*L227-238: Which same method is applied to the radiosonde data? Please provide more information on the conducted analysis and its results within the method and results section.*

Here we were referring to the methodology to identify below cloud decoupling layers from the equivalent potential temperature profile. Following your suggestion we have added the details of this analysis to both the methods and results section.

See lines 173 to 176 in the methods:

"To contextualise the results of this study with respect to the longer term dataset of cloud and boundary layer structure at Summit, we also apply the same methodology to detect decoupling layers from 932 Vaisala radiosonde profiles launched at Summit during cloudy conditions in June, July, and August between 2010 and 2022 (Shupe and Walden, 2010). Where 'cloudy' profiles are identified from the positive detection of a cloud base height by the ceilometer (Shupe, 2010)."

And lines 240 to 243 in results:

"Considering all the June, July, and August radiosonde profiles during cloudy times at Summit from 2010 to 2022 (932 radiosonde profiles in total), below-cloud thermodynamic decoupling layers occur 49% of the time. The average lowest decoupling height in this dataset was 120 m and 90% of decoupling heights were below 250 m."

And modified lines 267 to 276 in the discussion:

"The longer term dataset (2010-2022) of radiosonde launches at Summit demonstrate that the surface layer is decoupled from the sub-cloud layer 49% of the time (Shupe and Walden, 2010), which is similar to observations from the central Arctic Ocean (Sotiropoulou et al., 2014; Brooks et al., 2017; Vüllers et al., 2021). This implies that during half of the summer period over the central Greenland Ice Sheet, when surface melt is significantly influenced by cloud properties (e.g. Bennartz et al., 2013), surface aerosol measurements do not accurately represent the aerosol population relevant to cloud interactions. Consequently, relying solely on surface aerosol measurements is inadequate for studying cloud-aerosol interactions in this context. In winter months, surface aerosol measurements are even less likely to be representative of the cloud relevant population, since persistent high static stability at the surface occurs over 80% of the time (Miller et al., 2013), and clouds with base heights < 2,000 m are less common (Shupe et al., 2013). This could explain the particularly low surface aerosol particle number concentrations at Summit Station during the winter (Guy et al., 2021)."

**Response to RC2:**

*Please provide description how your setup was calibrated, the POPS has own issues as mentioned by other reviewer.*

Please see the detailed response to RC1 regarding the POPS calibration, uncertainty analysis and limitations. We have added more information in section 2.3 to clarify this for the reader, see lines 95 to 123 (also quoted in the response to RC1).

*Also any aerosol and meteo data from surface measurements would strengthen the manuscript, put your measurements into context of surface observations. Currently, the data is rather providing relative changes in vertical domain, any intercomparison to surface observation would help the reader to orientate what part of aerosol distribution is covered by POPS measurements. Were you observing particle growth after new particle formation, plums that are not observed at surface, long range transport, etc...?*

Unfortunately, due to multiple maintenance projects that were ongoing at Summit during our campaign, the availability of coincident surface-based measurements was limited. Surface turbulent flux measurements and the millimetre cloud radar were offline for the duration of the campaign. The only size resolved surface aerosol measurements at Summit are those from another POPS instrument, so it is not possible to investigate the wider aerosol distribution. We did compare the profile measurements with those from the surface-based POPS, and at your suggestion we have now included this in the manuscript as an additional figure (figure 4 in the revised manuscript). This figure demonstrates that the changes observed in the vertical profile observed in cases (d) to (f) were indeed a function of height rather than temporal changes.

*Please provide details on aerosol sampling setup, inlet (heated/non-heated), length, diameter, expected losses. Were there any corrections applied? What particle diameter you provide? Is it PSL equivalent or any corrections based on known refractive index of particles?*

See full details in the response to RC1.

*Provide the uncertainties of all sensors and their operation range.*

The uncertainties in the POPS measurements are discussed in more detail in response to RC1 above. We have now added the combined uncertainties from the manufacturers sampling efficiency estimation and flow-rate calibration, and the noise from the onsite zero check to the published POPS data files and have included error bars illustrating these uncertainties on figures 4 and 6. Additional uncertainties related to the refractive index of the PSL spheres, and possible calibration drift (which may particularly impact the AWO-POPS that has been operating at Summit for one years) are not quantified by have now been pointed out in the text (see lines 105 and 111). The upper limit of detection for the POPS is 10,000 particles per second, which is far above the concentrations that we measure at Summit.

The uncertainties in the Windsond sensors are stated in section 2 ($\pm$1 hPa, 0.3 $\circ$C, and 2 % relative humidity). The operational range of the Windsond is -40 to 80 °C and 0 to 100% RH.